# Validation of the Internal Coherence Scale (ICS) in Healthy Geriatric Individuals and Patients Suffering from Diabetes Mellitus Type 2 and Cancer

**DOI:** 10.3390/geriatrics9030063

**Published:** 2024-05-14

**Authors:** Annette Mehl, Anne-Kathrin Klaus, Marcus Reif, Daniela Rodrigues Recchia, Roland Zerm, Thomas Ostermann, Benno Brinkhaus, Matthias Kröz

**Affiliations:** 1Research Institute Havelhöhe, Kladower Damm 221, 14089 Berlin, Germany; anne-kathrin-klaus@posteo.de (A.-K.K.); roland.zerm@havelhoehe.de (R.Z.); matthias.kroez@klinik-arlesheim.ch (M.K.); 2Society for Clinical Research, Hardenbergstraße 20, 10623 Berlin, Germany; marcus.reif@gkf-berlin.de; 3Lehrstuhl für Forschungsmethodik und Statistik in der Psychologie, University Witten/Herdecke, 58455 Witten, Germany; daniela.rodriguesrecchia@uni-wh.de (D.R.R.); thomas.ostermann@uni-wh.de (T.O.); 4Department of Internal Medicine, Havelhöhe Hospital, Kladower Damm 221, 14089 Berlin, Germany; 5Institute for Integrative Medicine, University Witten/Herdecke, Gerhard Kienle Weg 8, 58313 Herdecke, Germany; 6Institute for Social Medicine, Epidemiology and Health Economics, Charité, Universitätsmedizin Berlin, Luisenstr. 57, 10117 Berlin, Germany; benno.brinkhaus@charite.de; 7Klinik Arlesheim, Research Department, Pfeffinger Weg 1, 4144 Arlesheim, Switzerland

**Keywords:** internal coherence, salutogenesis, geriatrics, validation

## Abstract

Background: With increased life expectancy, the coexistence of functional impairment and multimorbidity can negatively impact life quality and coherence in geriatric individuals. The self-report 10-item Internal Coherence (ICS) measures how individuals cope with and make sense of disease-specific life challenges. The aim of this study was to validate the ICS in a sample of geriatric individuals. Methods and Procedure: In a cross-sectional study, geriatric individuals with and without chronic diseases were recruited. A factor analysis with principal component extraction (PCA) and a structural equation model (SEM) was conducted to assess the ICS factor structure in a geriatric sample. To measure convergent validity, the following scales were used: Short Health Survey (SF-12), Karnofsky Performance Index (KPI), Trait autonomic regulation (Trait aR), Sense of Coherence Scale (SOC), and Geriatric Depression Scale (GDS). Results: A sample of *n* = 104 (70–96 years of age) patients with Diabetes Mellitus Type 2 (*n* = 22), cancer diseases (*n* = 31) and healthy controls (*n* = 51) completed the ICS. PCA and SEM yielded the original two-factor solution: 1. Inner resilience and coherence and 2. Thermo coherence. Overall internal consistency for this cohort was satisfying (Cronbach’s α with r_α_ = 0.72), and test-retest reliability was moderate (*r_rt_* = 0.53). ICS scores were significantly correlated to all convergent criteria ranging between *r* = 0.22 * and 0.49 ** (*p* < 0.05 *; *p* < 0.01 **). Conclusion: Study results suggest that the ICS appears to be a reliable and valid tool to measure internal coherence in a geriatric cohort (70–96 years). However, moderate test-retest reliability prompts the consideration of potential age-effects that may bias the reliability for this specific cohort.

## 1. Introduction

Advancement and increased accessibility in health care contributed to a steadily rising geriatric population ranging from 80 to 100 years in Germany, with one in five people older than 65 years [1]. In 2021, 7 of 100 people passed the age of 80 years and represent the fastest growing proportion across all age cohorts [2]. Conversely, as life expectancy increases, older people suffer longer or later in life from chronic somatic conditions (e.g., coronary heart disease, Diabetes Mellitus Type 2 and/or cancer), co- and multimorbidity and functional limitations [3]. Recent studies among the elderly show that multimorbidity in this population increases the risk of polypharmacy [4], the use of addictive medication such as neuroleptics and benzodiazepine [5], and the likelihood to experience depression [6,7], anxiety and stress [8]. Additionally, multimorbidity can lead to feelings of elevated interpersonal dependency [9] and poor health-related quality of life (HRQL) [10]. This evidence implies the necessity of research focusing on resources establishing physical and mental health in elderly individuals [11].

According to Kuhlmey [12], the restoration of health and HRQL in multimorbid patients is non-hierarchical and requires specific adaptive physiological [13,14] and psychosocial skills for mental health [11,15]. Although the clinical assessment of HRQL in the geriatric population gained considerable importance in the last two decades [16,17], there is still a lack of instruments that measure adaptive physiological and psychological capabilities. Such instruments should go beyond a dichotomous determination of health and disease, and should be tailored to adaptive capacity, resilience and coping [18,19]. Two concepts that—when combined—capture the physiological adaptation of autonomic functions and the process of successful psychological coping to achieve meaning in life are (a) the hygiogenesis model, developed by researchers as Hildebrandt and colleagues [13,14,20,21], and (b) Antonovsky’s salutogenesis model [15]. Hildebrandt’s concept of hygiogenesis can be characterized as the auto-regulative physiological self-healing processes of the organism, fostered by therapeutic stimulation. It contains the physiological degree of functional rhythmic adaptation and functional normalization [22]. The concept of autonomic regulation (aR) captures autonomic functions and attempts to operationalize the concept of hygiogenesis [23], available as Trait- and state-based self-report scales (Trait aR and state aR). The Trait aR measures autonomic functioning via three dimensions: (1) Orthostatic circulatory regulation, (2) Rest/activity regulation, and (3) Digestive regulation [24]. Four dimensions with an additional thermoregulation factor were extracted for the state aR [25]. 

The salutogenesis model is a psychosocial resource-oriented perspective on health, focusing on variables that keep a person healthy. The heart of the salutogenesis model is the three-faceted central element “Sense of Coherence” (SOC), which is defined as an individual’s global orientation to consider life as understandable (comprehensibility), manageable (manageability) and meaningful (meaningfulness) [11,15,26]. Several studies show that individuals with a strong SOC are able to clarify and structure the nature of stressors [27] and additionally make appropriate use of general resistance resources [28,29]. SOC is a predictor for less morbidity and mortality and predictive of cancer survival [30]. 

The long version of the SOC self-report (29-item) as well as the short 13-item scale [31,32] have been repeatedly criticized owing to the supposed—yet not replicated—three-factor solution [33,34]. Another conceptual criticism is that SOC is considered as a past-oriented trait. An alternative to the SOC scales is the Internal Coherence Scale (ICS) [35] which is present- and future-oriented, and specifically developed for internal medicine and oncological patients to capture inner coherence and resilience. Internal Coherence is the combination of inner resilience, coherence, and thermo coherence. Coherence is described as an inner ability to adapt to challenges in life and to experience them as meaningful with a feeling of thermal comfort. The ICS shows a stable two-factor structure with robust reliability for individuals, aged 30–83 years, but lacks a validated version for a geriatric cohort aged 84 and older. Since salutogenetic- and hygiogenetic-oriented questions are particularly relevant, but have yet to be adequately investigated in geriatric individuals, the goal of this study, which was conducted in the context of a dissertation, was to validate the ICS in a geriatric sample of oncology patients, patients with Diabetes Mellitus Type 2, and healthy controls [36]. 

## 2. Methods and Procedure

### 2.1. Ethics and Framework of the Study

This cross-sectional study was conducted at the Gemeinschaftskrankenhaus Havelhöhe, Berlin, in three retirement homes of the Volkssolidarität, Berlin, the Johannesstift Öschelbronn and the Cusanus-Haus Stuttgart-Birkach, Baden-Württemberg, Germany. Additionally, participants from six general practitioners in Berlin were included from December 2013 to June 2016. The study was part of a medical dissertation published in 2021 at the Charité Berlin. The study was operated according to the Declaration of Helsinki Guidelines, and approved by the local ethics committees at the Charité Berlin, as well as the Ethics Committee in Baden-Württemberg (application number EA1/258/13). Additionally, the study was subject to on-site monitoring. All participants read the study information and provided written informed consent [36]. 

### 2.2. Participants, Inclusion and Exclusion Criteria

Participants were recruited via flyers in the sport and leisure sector and via medical contacts. Based on the different study groups (healthy participants and individuals with two internal medical conditions: oncology patients and patients with Diabetes Mellitus Type 2) we conducted a Power Analysis (PA) with the power set at 90%. The PA resulted in a sample of *n* = 69 participants across the study groups. Considering a drop-out rate of 10%, the calculated sample size was increased to 78 participants. Inclusion and exclusion criteria are displayed in Table 1.

### 2.3. Self-Report Questionnaires

Participants were asked to complete the following self-report questionnaires. The data of the Internal Coherence Scale were used for the validation study. Data of all other questionnaires were used as convergence criteria to assess convergent validity for the ICS questionnaire. 

### 2.4. Internal Coherence Scale (ICS)

The 10-item ICS [35] comprises two subscales: (1) Inner resilience and coherence and (2) Thermo coherence. The short self-report scale consists of a five-point ordinal scale that reaches from 10–50. It reveals good to very good reliability with Cronbach’s α with *R_α_* = 0.91 and a test-retest reliability of *r_rt_* = 0.80. It was validated for healthy individuals, oncological and internal medicine patients and patients with mental illness aged 18–83 years (Kröz et al., 2009) [41]. In terms of external validity, it showed satisfying to good reliability with the SOC, *r* = 0.43–0.72 (*p* < 0.001).

### 2.5. Trait Autonomic Regulation (Trait aR) 

The Trait aR [24] measures autonomic functions with 18 items on a 5-point Likert scale ranging from 18–54 for the overall score. It reveals satisfactory internal consistency with Cronbach’s α with *R_α_* = 0.75 and good test-retest reliability with *r_rt_* = 0.85 for the age group from 18–85 years. The Trait aR captures the construct via 3 subscales: (1) Orthostatic circulatory regulation, (2) Rest/Activity regulation, and (3) Digestive regulation [24] 

### 2.6. The Sense of Coherence Scale (SOC-13)

The SOC-13 was originally created by Aaron Antonovsky and further developed by several working groups [15,31,42,43]. It captures SOC via three theoretical components: comprehensibility, manageability, and meaningfulness. The SOC 13 is 5-point Likert scaled and demonstrated satisfactory internal consistency in a German sample (aged 19–92 years) with Cronbach’s Alpha *R_α_* = 0.85 [26,44] and *R_α_* = 0.77 for the geriatric group (85–95 years) [45]. 

### 2.7. Short Form Health Survey (SF-12) and Karnofsky Performance Index (KPI)

The SF-12 [46] is a short generic self-report measure to assess patients’ health-related quality of life (HRQL). The 12 items form a physical component score (PCS) and a mental component score (MCS) covering physical functioning, pain, general health perception, vitality, psychological well-being, social functioning and emotional role function. Both components showed satisfying reliability for the PCS and MCS and an overall Cronbach’s alpha of 0.77 [47]. The Karnofsky Index (KPI) [38] is a peer-review instrument that measures physical functioning in daily life. This physician-assessed indicator measures physical functioning in daily life in 10% steps with a range of 0% (dead)–100% (normal). 

### 2.8. Geriatric Depression Scale (GDS-15)

The GDS, originally designed by Yesavage and Sheikh as a 30-item self-report questionnaire [48], is also available as a 15-item screening tool (GDS-15) [49] and detects depression in older adults [50]. The GDS uses a *“yes”* or *“no”* format that sums up to 15 points (5 < no depression, 5–9 = mild/moderate depression, 10 ≥ severe depression). Internal consistency for the 15-item screening tool (*R_α_* = 0.88) and the correlation between the long and short version (*r* = 0.89) is high among inpatients [50]. Test-retest reliability ranges between *r_rt_* = 0.68 and 0.85 across international studies, e.g., [51,52].

## 3. Statistical Procedure

The demographic description of the sample consisted of three groups (oncology patients, patients with Diabetes Mellitus Type 2, and healthy controls), and it was displayed in relative and absolute frequencies or means and standard deviations for categorical- or interval-scaled variables, respectively. The scale scores of self-report questionnaires for the participants were omitted when 20% or more of the items were missing values. Differences in social demographics for the groups were calculated using nonparametric test procedures (e.g., chi-square or Kruskal–Wallis tests for categorical or interval data to discriminate between the groups). The validation procedure of the ICS was conducted in a three-step statistical procedure. First, the factor structure of the ICS was assessed, using a principal component extraction (PCR) with varimax rotation of the original ICS version [35]. Second, to confirm the results of the PCR, a structural equation model (SEM) (generalized multivariate regression model) was calculated to evaluate the “goodness-of-fit” of the geriatric ICS data structure. Third, overall and subscale reliability (internal consistency) was calculated using a reliability analysis (RA), resulting in Cronbach’s alpha indices, scores for ‘alpha if item deleted’, and item total correlations. Bivariate Spearman’s partial rank correlations, stratified for the three study groups, were performed for the patient reported outcomes to (a) exclude pseudo-correlations between the three study groups (healthy participants, oncological patients and patients with Diabetes Mellitus Type 2) and (b) to examine test-retest reliability using the sum score from baseline and two-week follow-up data. Lastly, to evaluate explorative group differences for validity, the Aligned Rank Test was used, stratified for the healthy control and the two patient groups. Therefore, the patient study groups were split into four subgroups due to large differences in disease duration time to detect potential subgroup differences. The oncology group was divided into: Long-term survivors > 5 years of survival and short-term survivors ≤ 5 years of survival. Patients with type 2 diabetes were grouped into: Long-term diabetes > 5 years of duration of sickness and short-term diabetes ≤ 5 years of duration of sickness. For statistical analyses the software packages SPSS Version 26 [53] and SAS 9.4 [54] were used. For the SEM analysis the software R [55] and the package Lavaan [56] were used.

## 4. Results

### 4.1. Participants

A total of *n* = 202 participants (healthy individuals, individuals with cancer diagnoses and Diabetes Mellitus Type 2) were initially recruited, with *n* = 104 meeting all inclusion criteria. N = 97 were excluded because they did not meet inclusion criteria or were no longer interested in the study. Among the *n* = 104 study participants, 27% were recruited from the sport and leisure sector, 32% from medical facilities, and 41% from their homes, predominantly in senior living facilities with assisted living. The validation study was conducted with *n* = 104 participants (*n* = 51 healthy controls; *n* = 31 individuals- with cancer diagnoses and *n* = 22 with Diabetes Mellitus Type 2). Notably, the 31 participants in the oncology group presented a range of different malignant tumor conditions (past or present): dermatologic carcinomas (*n* = 10), breast carcinomas (*n* = 6), colon/rectal carcinoma (*n* = 5), uterine/cervical/ovarian carcinoma (*n* = 5), prostate carcinoma (*n* = 3), neuroendocrine tumor (*n* = 1), myelodysplastic syndrome (*n* = 1), small bowel tumor (*n* = 1), and ENT carcinoma (*n* = 1). The average recurrence-free period showed a mean of M = 12.03 and a standard deviation of SD = 12.02 years, with a median of 10 years and a range from 0 to 41 years. Appendix A shows the demographic characteristics of *n* = 104 participants taking part in the study. All *n* =104 participants provided informed consent and completed the Internal Coherence Scale (ICS) [35] for validation purposes, and additional other self-report questionnaires, which are described below [36].

### 4.2. PCA and Structural Equation Model (SEM)

The results of the PCA identified a two-factor solution which explained 59.22% of the total variance. The factor analysis (Kaiser–Meyer–Olkin: KMO = 0.70; Bartlett Test; *p* = 0.00) revealed the identical two-factor solution of the original version with subscale 1: Inner resilience and coherence (8 items) and subscale 2: Thermo coherence (2 items). For a detailed description of the PCA results see Appendix A. To confirm the two-factor solution of the ICS, a structural equation model (SEM) was calculated, which yielded a good data fit with robust reliability of the two subscale structure solution with the comparative fit index (CFI) 1.00 (recommendation > 0.95); Tucker–Lewis index (TLI) 1.00 (recommendation > 0.95); Root Mean Square Error of Approximation (RMSEA) < 0.001 (recommendation < 0.05) and Standardized Root Mean Square Residual (SRMR) of 0.03 (recommendation < 0.05). The inner consistency yield satisfying the reliability of the ICS and the two subscales with item-total correlation are displayed in Appendix A. With the SEM we confirmed the two-factor structure of the ICS. Individual group comparisons of ICS scores (sum score, inner resilience and coherence scale, and thermo coherence) between groups and subgroups are specified in Appendix A). Bivariate Spearman’s partial rank correlations, stratified for the three study groups, showed that higher Internal Coherence was associated with higher HRQL (SF-12), KPI and SOC, but lower GDS (detailed correlations in Appendix A).

## 5. Discussion

In this cross-sectional validation study the factor structure of the original version of the Inner Coherence Scale (ICS) with two subscales: 1. Inner resilience and coherence and 2. Thermo coherence [35] could be replicated for this geriatric cohort (aged 84–96 years). In addition, the ICS measures inner coherence and resilience and thermo coherence with sufficient reliability. Comparing our results with two past ICS validation studies that examined younger samples with a mixed age range between 30–83 years [35] and 19–74 years [41], the two-factorial solutions were identical in this elderly cohort aged 84–96 years, confirmed by a PCA and a subsequent confirmative SEM. An additional reliability analysis also showed satisfactory internal consistency and moderate test-retest reliability for elderly individuals. Compared to the original ICS version [35], and the study by Trapp (2014), the internal consistency for the global score and subscale 1 (inner resilience and coherence) was less homogenous with r_α_ = 0.72, for the sum scale compared to r_α_ = 0.91 for the original ICS validation. In contrast, the study by Trapp (2014) showed lower overall internal consistency, caused by more variability in the item responses of subscale 2 (thermo coherence). Additionally, test-retest reliability was lower in the geriatric group compared to former ICS validation studies including younger individuals. This points to more time-related variability in elderly individuals [35]. The lower internal consistency of the ICS displayed in this age cohort may indicate an age effect. With increasing age, item inter-correlations, especially for the “inner resilience and coherence subscale” (subscale 1), dissolved. Reasons for lower correlations are the prospect of a shorter remaining lifespan, combined with simultaneously experiencing multimorbidity and functional limitations, which can lead to the use of polypharmacy [4] and prolonged addictive medication [5]. In a recent study, polypharmacy has been found to be strongly associated with disease burden in geriatric individuals [5]. It is likely that the prospect of not being able to restore health negatively reflects on coherence and less stable ICS scores. The capacity of adaptation and resources are subject to change throughout the life span, and tend to diminish with aging, e.g., [57,58]. The reasons are manifold, with more physical restraints caused by less vitality and problems in everyday life, and reduced functioning due to old age. This is in line with the findings when comparing the item loadings for subscale 1 of the current validation and the original validation. The highest factor loading deltas were for the item “feeling secure” and the item “feeling confident”. For these two items, item total correlations were lowest, resulting in a greater variability of item responses and lower Cronbach Alpha scores for subscale 1. Besides the greater variability in subscale 1 item responsiveness, the overall stability of the ICS scores decreased over time, reflecting in a lower test-retest reliability with *r* = 0.53 (*p* < 0.01) when compared to the original validation study with *r* = 0.80 (*p* < 0.05). It is likely that, similar to metastasized cancer patients, geriatric patients also have a greater variability representing “good” and “bad” days [59]. In contrast, a sample of breast cancer patients with cancer-related fatigue showed significantly improved ICS scores at a 6 month follow-up when treated with a 10 week multimodal anthroposophic therapy program, compared to a group receiving standard aerobic training [60], indicating that “coherence” is a trait to be manipulated rather than a static trait like SOC. Similarly, a study by Oei et al. (2019) showed improved ICS scores in a group of breast cancer patients who received supportive viscum album treatment [61]. 

### 5.1. Correlation and Subgroup Analysis

In terms of ICS correlations with external criteria regarding the entire group, low Internal Coherence was correlated with lower education and less exercise. Higher Internal Coherence correlated with better HRQL (SF-12), higher KPI, higher SOC, Trait aR and lower GDS scores. A detailed explorative analysis of the oncology survivors and diabetes patients showed several significant subgroup differences regarding the ICS sum score and the subscale score of inner resilience and coherence using explorative nonparametric stratified Aligned Rank Tests (Appendix A). The results showed significant higher ICS sum scores for long-term oncology survivors compared to healthy controls and long-term diabetes patients. Additionally, for the inner resilience and coherence scores (ICS subscale 1), significantly better ICS resilience scores for long-term oncology survivors compared to healthy controls and long-term diabetes patients were found; and the superiority of short-term diabetes patients compared to long-term diabetes patients was found. These results are in line with a study by Márquez-Palacios et al. (2020), indicating that coherence (measured in this study as SOC) has a strong correlation with diabetes in different phases of the disease [62]. Additionally, SOC seems to be a protective factor for demoralization regarding women with a recent gynecological cancer diagnoses [11,63], and long-term cancer survivors with high SOC, in which a strong SOC predicted survival in a Hawaiian sample [64]. 

### 5.2. Limitation and Strengths

We also want to report on the study limitations. The first is methodological, referring to a potential selection/recruitment bias regarding the overall group and, specifically, the oncological group. Most of the study participants from this group were recurrence-free cancer survivors, of which *n* = 19 participants had been free of recurrence for more than five years. Accordingly, long-term survivors (>5 years) showed higher ICS scores than short-term survivors (≤5 years) (Appendix A). A second limitation might stem from another selection bias in regards to the various cancer diagnosis (displaying, e.g., a high amount of skin tumors) and long cancer recurrence-free time [36]. Other study limitations concern the sociodemographic variables (e.g., education or living with a partner), which were also positively correlated with ICS scores. Our sample consisted of older-aged individuals who were highly educated and reported high quality of life, which could have impacted coherence and resilience. Our sample would have benefitted from controlling demographic variables during the statistical procedure. However, this study is the first that examines Internal Coherence in a geriatric sample using a sound statistical procedure. In addition, it benefits from a diverse sample that consisted of three subgroups: oncological patients and survivors, patients with Diabetes Mellitus Type 2, and healthy controls, which strengthens the overall reliability of the validation and the self-report questionnaire. The solid methodological approach, containing both a PCA and a subsequent structural equation model to confirm the results of the PCA, is especially beneficial for the study results.

## 6. Conclusions

The ICS is the first validated self-report questionnaire to reliably measure inner resilience, coherence and thermo coherence. Study results suggest that the ICS appears to be a reliable and valid tool to measure Internal Coherence for an older-aged cohort as well. Moderate test-retest reliability prompts consideration of potential age effects that may bias reliability in this elderly cohort. Further research has to be conducted to better understand how Internal Coherence develops across the lifespan and how it can be improved in the elderly population. 

## Figures and Tables

**Table 1 geriatrics-09-00063-t001:** Inclusion and Exclusion Criteria for study groups.

	Inclusion Criteria (a–f)	Exclusion Criteria (a–h)
Oncological Group	Age ≥ 70 years Mobility activity level 1 or 2 according to Siegmar et al. (1982) [37], corresponding to at least walking independently with or without assistive device Karnofsky performance index > 50 [38] Malignant disease; current manifested or in history.	Diabetes Mellitus (Type 1/2) Neurological disease (e.g., stroke) Psychosis Cognitive impairment (Mini Mental Status Test < 12/20, MMST SF) [39] Tumor specific surgery Drug therapy or radiotherapy in the last 4 weeks
Diabetes Group	see oncological group: a–c e. Diabetes Mellitus Type 2	see oncological group: b–d g. Malignant disease currently manifested or in history
Healthy group	see oncological group a–c f. Organic illness severity < 3 according to Cumulative Illness Rating Scale (CIRS): 0–4 in the range of visual and auditory impairment) [40]	see oncological group a–d, g h. Organic disease severity ≥ 3 according to CIRS (except visual and hearing impairments).

## Data Availability

Any request for data or materials should be addressed in written form to the corresponding authors.

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
