# Peer review of "Validation of the Internal Coherence Scale (ICS) in Healthy Geriatric Individuals and Patients Suffering from Diabetes Mellitus Type 2 and Cancer"

_geriatrics, 2024, doi:10.3390/geriatrics9030063_

Round 1

Reviewer 1 Report

Comments and Suggestions for Authors

Thank you for your submission. I think there is some work that is needed in order to make the submission publishable in the journal. I have provided handwritten comments on your paper & have attached to be sent to you.

Some of the issues:

- You have mixed APA & AMA formatting of the reference list. You need to follow the journal required formatting. The slash lines at the numbers mean the article was sited in the body of the paper. There are several citations that are not in the reference list. There is 1 reference in the list that I did not find cited (I may have missed it.

- Some of the references are VERY old, especially those related to reporting data on the older adult population. I understand some of these articles are considered "classics", but there is newer information available. Up to date is considered within the past 5 years.

- Much of your reference list is in the German language & needs to be changed to English.

- There are many words in the text that need to be defined for the reader. Readers need to be educated more about your topic, including what words mean. For example, hygiogenesis, salutogenesis, coherence, etc. If you do not define words - when they are mentioned - some people will not read the article. This should be an easy task to accomplish.

- You used data from 2013-16 study - making this old data. You need to explain more than that this was a dissertation. Why is the data still pertinent? Has nothing been done on the topic since 2016? Why would the reader be interested in reading this?

- You write in the 1st person ("we", "our") which is generally not acceptable. You need to check with the journal editor. Using "the authors", "the researchers", "the research team", etc. are acceptable substitutes.

- You use symbols without explaining them. While they are common & familiar to some, they might not be familiar to all readers. Therefore, write out the word & then present the symbol. This only has to be done once. For example, ... results were greater than (>) ... etc.

- Some of the information would be better presented if you put it in boxes, bullet point lists or even tables. One example would be all of the inclusion & exclusion criteria for the different components. This would greatly help the flow of the information.

- Every time you present statistics there must always be a zero (0) in front of a decimal point. For example - 0.272. This is a big issue in your tables & needs to be fixed.

- The flow of the paper should be from broad to narrowed information that is informational & follows a pattern of understanding. I would suggest creating an outline of a logical flow (including appropriate headings) for the content & then rearranging your information so there is a clear beginning to a clear ending.

I hope you will accept this feedback & make the revisions. Best Wishes!

Comments on the Quality of English Language

Some of the issues are: grammar, punctuation, tense, sentence structure.

Much of the reference list is in the German language.

I have attached a copy of your submission with my handwritten comments. If you follow my comments, I have offered suggestions for improving the issues mentioned above. 

Reviewer 2 Report

Comments and Suggestions for Authors

Validation of the Internal Coherence Scale (ICS) in healthy geriatric individuals and -patients suffering from Diabetes Mellitus Type 2 and Cancer

Thank you very much for allowing me to review the manuscript. I would like to make a series of suggestions regarding its publication.

Title: The title adequately reflects the manuscript.

Introduction:

- Review the bibliographic references to find some more updated ones. There are many references over 15 years old (do not consider classical references).

Methodology:

- Since data collection was carried out between 2013 and 2016, I am concerned that the data may not be suitable for the study as almost 10 years have passed and societal characteristics may have changed.

- In the "ethics & framework of the study" section, the methodology followed should be indicated, as it is a cross-sectional study and is indicated in the abstract.

- Include how the sample size calculation was performed.

- It is not adequately justified why the sample has been separated into three groups depending on the pathologies and what the inclusion of a control group entails. Could you explain this to me?

- Ceiling and floor effects should be measured.

- To validate the instrument, model fit measures should be evaluated and confirmatory factor analysis conducted.

- The Bartlett's test and KMO analysis are missing.

Results:

- The items in Table 2 are disordered, and it is difficult to read the numbers. I recommend justifying the text to the left.

- I am unable to see if the second objective has been adequately addressed in the results. I believe this needs to be fixed.

Round 2

Reviewer 2 Report

Comments and Suggestions for Authors

Dear authors, thank you very much for allowing me to review your manuscript again. I am aware of the effort you have made to incorporate all the comments and suggestions. The authors have addressed all the issues raised.

Author Response

Dear Reviewer, 

thank you very much on the careful revision on our manuscript. Attached is our revised manuscript

According to your revision request we changed the following references: Reference 1 and 2 are references of the German official Statistical data base for Geriatrics. Unfortunately we could not translate these references into English. We deleted however the following German reference:  3 and 14.  

As required, we furthermore checked the text for punctuation, grammar and sentence structure. 

We hope that our revision will now meet the requirements of your journal.  Thank you very much. We are looking forward to the decision on our paper. 

Sincerely yours, 

Dr. Annette Mehl
